# Survival in Colon, Rectal and Small Intestinal Cancers in the Nordic Countries through a Half Century

**DOI:** 10.3390/cancers15030991

**Published:** 2023-02-03

**Authors:** Filip Tichanek, Asta Försti, Vaclav Liska, Akseli Hemminki, Kari Hemminki

**Affiliations:** 1Biomedical Center, Faculty of Medicine, Charles University Pilsen, 30605 Pilsen, Czech Republic; 2Institute of Pathological Physiology, Faculty of Medicine in Pilsen, Charles University, 32300 Pilsen, Czech Republic; 3Hopp Children’s Cancer Center (KiTZ), 69120 Heidelberg, Germany; 4Division of Pediatric Neurooncology, German Cancer Research Center (DKFZ), German Cancer Consortium (DKTK), 69210 Heidelberg, Germany; 5Department of Surgery, School of Medicine in Pilsen, University Hospital, 30605 Pilsen, Czech Republic; 6Cancer Gene Therapy Group, Translational Immunology Research Program, University of Helsinki, 00290 Helsinki, Finland; 7Comprehensive Cancer Center, Helsinki University Hospital, 00290 Helsinki, Finland; 8Division of Cancer Epidemiology, German Cancer Research Center (DKFZ), Im Neuenheimer Feld 580, 69120 Heidelberg, Germany

**Keywords:** incidence, prognosis, relative survival, treatment, risk factors

## Abstract

**Simple Summary:**

Survival in colon and rectal cancers has internationally developed well, although reliable data span essentially only two decades. For small intestinal cancer, fewer data are available but survival appears to be improving. Overall, the exact causes of survival improvements are not known. During the 50-year period, 5-year survival in colon and rectal cancers improved linearly in Norway, while in Finland and Sweden the rate of improvement decreased with time, which is the opposite to Denmark where the rate increased. In small intestinal cancers, the rate of improvement was linear or increasing. The remarkable Danish achievement of improving relative survival rates more than the other counties coincided with cancer policy planning and instigating economically backed organizational and infrastructural improvements. The slowing survival rates in the other countries call for optimization of the available resources and a search for novel approaches.

**Abstract:**

**Background:** Survival studies in intestinal cancers have generally shown favorable development, but few studies have been able to pinpoint the timing of the changes in survival over an extended period. Here, we compared the relative survival rates for colon, rectal and small intestinal cancers from Denmark (DK), Finland (FI), Norway (NO) and Sweden (SE). **Design:** Relative 1-, 5- and 5/1-year conditional survival data were obtained from the NORDCAN database for the years 1971–2020. **Results:** The 50-year survival patterns were country-specific. For colon and rectal cancers, the slopes of survival curves bended upwards for DK, were almost linear for NO and bended downwards for FI and SE; 5-year survival was the highest in DK. Survival in small intestinal cancer was initially below colon and rectal cancers but in FI and NO it caught up toward the end of the follow-up. **Conclusions:** Relative survival in intestinal cancers has developed well in the Nordic countries, and DK is an example of a country which in 20 years was able to achieve excellent survival rates in colon and rectal cancers. In the other countries, the increase in survival curves for colon and rectal cancer has slowed down, which may be a challenge posed by metastatic cancers.

## 1. Introduction

Colon cancer is in men about 1.5 times and in women 2 times more common than rectal cancer and over 10 times more common than small intestinal cancer (SIC) in Northern Europe [1]. These cancers are largely adenocarcinomas, except that in SIC adenocarcinomas and indolent neuroendocrine tumors (NET, carcinoids) they are present with approximately equal proportions [2,3,4,5,6]. SIC also includes rare sarcomas and gastrointestinal stromal tumors. The earlier increase in the incidence of colon and rectal cancers has slowed down and mortality has started to decline in all Nordic countries; the contribution of the organized screening to mortality decline is unclear, as it was implemented only in Finland (FI, guaiac-based fecal occult blood test in 2004–2016 followed by fecal immunohistochemical testing, FIT) and in Denmark (DK, FIT since 2014), and not in Norway (NO) or Sweden (SE) [7]. The incidence of SIC has continuously increased in DK and SE, particularly for adenocarcinoma; improving imaging techniques have most likely contributed to the detection of NETs [5,6]. Colon and rectal cancers are often considered jointly as colorectal cancer (CRC) as they share many aspects of biology and pathology, including risk factors which include many dietary factors, physical inactivity, obesity, type 2 diabetes, smoking, alcohol and inflammatory bowel disease [8,9,10]. Many of the risk factors are also shared in SIC, but smoking and alcohol may be exceptions [11,12,13,14,15]. Family history is a feature of all of these cancers, and familial risk for SIC, particularly for NET, is one of the highest among all cancers [16,17,18,19,20]. All of these cancers are manifestations in Lynch syndrome, familial adenomatous polyposis and Peutz–Jeghers syndrome [2,11,21].

International treatment guidelines for these cancers are followed in the Nordic countries and, additionally, national guidelines have been developed with details for SE as described elsewhere [22]. Surgery is the main treatment modality, often featuring segmental resection, frequently in combination with chemotherapy, when risk factors such as nodal dissemination are present. Chemotherapy typically includes 5-fluorouracil/leucovorin, as an infusion or as an oral prodrug, in combination with oxaliplatin or topoisomerase I inhibitor irinotecan [8,23]. In rectal cancer, radiation is almost always given before or after surgery. Immunotherapies show promising results, particularly in patients with hereditary (Lynch syndrome) or sporadic mismatch repair defects [8,23]. While diagnostic endoscopy has become commonplace for colon and rectal cancers, it has not yet become commonplace for SIC. NET is an indolent tumor which may be surgically removed [2,5,24]. NET may present as multiple synchronous or asynchronous primaries, but it rarely metastasizes with the primary target being the liver [18,25,26]. Global survival in these cancers has developed favorably [5,22,27,28].

We aimed to investigate the long-term survival trends in intestinal cancers in the Nordic countries using the NORDCAN database, with a focus on the timing of changes in relative survival and the possible reasons for these over a 50-year period from 1971 to 2020. We mapped trend changes via breakpoint analysis in 1- and 5-year survival and in conditional 1- to 5-year (5/1-year) survival. For mapping detailed survival changes, we show a novel measure of the estimated change in survival over a year (annual change). We compare survival rates in DK, FI, NO and SE. The countries have historical and cultural ties and share the same basic medical care organization, but economic health care resources may influence their delivery (www.macrotrends.net). All Nordic countries have been offering medical care practically free-of-charge to the population. Thus, the present results describe a ‘real world’ experience of medical outcomes through a half century. As changes in cancer incidence/mortality influence relative survival rates, we also report the background incidence and mortality rates in these cancers over the same period [29].

## 2. Methods

The data used originate from the NORDCAN database 2.0 which is a compilation of data from the Nordic cancer registries [30]. These registries are presented in detail by Pukkala and coworkers [31]. The database was accessed via the International Agency for Cancer (IARC) website (https://nordcan.iarc.fr/en). The coverage of cancers in these cancer registries is generally considered high [31]. The SE cancer registry does not consider cancers in death notifications, and some 4% of cases may be missed because of this; an overall comparison of various health records showed that the coverage was over 90% [31,32]. The comparability of diagnostics over a 50-year period may be an issue. The SE cancer registry used ICD-7 from the start of registration in 1958. When new codes have been taken to be used, all diagnoses are additionally recorded in the ICD-7 system to maintain consistency [33]. FI uses a code conversion system to maintain consistency.

Data on colon and rectal cancers and SIC were extracted from NORDCAN and the follow-up was extended until death, emigration or loss of follow-up, or to the end of 2020. In incidence and mortality analyses for age standardization, the world standard population was used. For incidence data, the starting date was either 1961 (the earliest available for all countries) or 1971 to match the earliest date for survival data. Survival data for relative survival were available from 1971 onwards and the analysis was based on the cohort survival method for the first nine 5-year periods, and a hybrid analysis combining period and cohort survival in the last period of 2016–2020, as detailed [34,35]. Age-standardized relative survival was estimated using the Pohar Perme estimator [36]. Age standardization was performed by weighting individual observations using external weights as is defined on the IARC website. Age groups 0 to 89 were considered. The DK, FI, NO and SE life tables were used to calculate the expected survival.

Statistical modeling and data visualizations were performed using R statistical software in the R studio environment (code available at https://github.com/filip-tichanek/nord_intestine). For a graphic presentation of incidence and mortality rates, lines were smoothed by the cubic smoothing spline using the R function ‘smooth.spline’ with a smoothing parameter (‘spar’) of 0.4 and with 15 knots.

Time trends of 1-year and 5-year relative survival (in %; obtained from NORDCAN for each of the 5-year periods) were modeled via Gaussian generalized additive models (GAMs) with thin plate splines (5 knots) and identity link. The GAM model included the effect of *group* (combination of sex and country) and group-specific non-linear effect of *time* (timepoint = middle year of each 5-year period) as predictors, allowing the estimation of the relative survival across a continuous time scale despite the discrete distribution of data points in time. As the input data (estimates of the 1-year and 5-year survival in each of the 5-year periods) were variously uncertain, standard errors for each data point (obtained from confidence intervals shown in the NORDCAN database) were included in the model. Models were run in the Bayesian framework using the ‘brms’ R package which employs ‘Stan’ software for probabilistic sampling. Separate models were used for different cancer localizations and 1-year and 5-year survival.

The prior distribution for the intercept was flat. The prior distribution for the effect of the *group* was defined to have a Gaussian distribution with zero mean and sigma of 20. We used Hamiltonian Monte Carlo sampling (2 chains, each of 7000 samples including 2000 warm-ups). All models were checked in terms of convergence, effective sample sizes and posterior predictive check.

For 5/1-year survival ratio estimation, we divided posterior draws from the 5-year survival model by posterior draws from the 1-year model to obtain the posterior distribution of the conditional survival and its estimated annual changes over time.

For all survival measures (relative 1-year and 5-year survival and 5/1-year ratio), we evaluated when the survival was changing over time with at least 95% plausibility (95% credible interval (CI) of the 1st derivation of given survival measure did not cross zero for at least 5 years). We also aimed to identify ‘breaking points’, i.e., times when the annual change in survival changed with at least 95% plausibility. This was assessed by calculation of the 2nd derivation of the given survival measure and its 95% CI; the ‘breaking point’ was defined as a peak value within at least a 3-year interval, where 95% CI for the 2nd derivation did not cross zero.

## 3. Results

### 3.1. Incidence and Mortality in the Nordic Countries

We show case numbers, age-standardized (world) incidence, mortality rates (ASR)/100,000 and cumulative risks for colon and rectal cancers and SIC in 2011–2020 in Table 1.

Age-standardized incidence trends for these cancers in men and women are shown from 1961 to 2020 in Appendix A. Male rates for colon cancer were initially close to the female rates but with time exceeding the female rates in all countries. While all rates increased over time, those for NO increased the most but appeared to have reached a maximum at around 2010. In the last period, NO and DK male rates were over 25/100,000, compared to FI women of 15/100,000. Incidence trends for rectal cancer were much higher in men compared to women (Appendix A). They declined modestly in DK, increased somewhat in FI and SE, and markedly increased in NO; the rates in the final period were 15/100,000 for NO and DK men, which is double that of FI and SE women. For SIC, incidence trends showed a strong upward bend before the year 2000 in all countries; in SE, the increase was modest (Appendix A). Male rates were higher than female ones with the maximal rate of 2.7/100,000 for NO men.

Mortality trends are shown in Appendix A. For SE, mortality started to decline after 1970: for DK and FI women it was after 1980 and for DK men it was after 1990. For NO, a steep decline started in 2010; for FI men, the mortality rate did not decline. For rectal cancer, mortality declined throughout the study period, except for NO men for whom it first increased and started to decline after the year 1990. Mortality rates for SIC showed fluctuation but overall modestly increased.

### 3.2. Relative Survival

Relative survival rates and their estimated annual changes are shown separately for each Nordic country (Figure 1, Figure 2, Figure 3 and Figure 4) and the underlying data for each survival metric are shown in Appendix A, pointing out periodic differences (by non-overlapping 95% CIs) and differences between the countries. In DK, male and female colon cancer survival curves were upward bending and 1- and 5/1-year curves were almost superimposable (Figure 1a,d). Significant upward breakpoints were observed for 1- and 5-year survival around the year 2000, and the positive development was confirmed by the solid lines for increasing annual changes; these were the highest for 5-year survival. Upward bending of the curves for rectal cancer and SIC were also apparent (Figure 1b,c,e,f). For rectal cancer, the increasing annual change showed a strong peaking at around the year 2000, marking the culmination of the increasing trend. Survival in SIC remained 10 or more % units below the other cancers. Annual changes for SIC are shown for all countries in Appendix A, but as SIC is a rare cancer, the fluctuations were large.

In FI, survival improvement in colon and rectal cancers showed a downward trend with declining annual changes (Figure 2a,b,d,e). For SIC, survival increased after some initial delay and 5-year survival matched that for colon and rectal cancers (Figure 2c,f).

For NO, colon cancer survival increased linearly and the annual changes remained relatively stable throughout the 50 years (Figure 3a,d). For rectal cancer, the survival curves showed a downward reflection (Figure 3b,e), while for SIC the increasing trends were maintained and 5-year survival reached the level of colon and rectal cancers (Figure 3c,d,f).

Survival curves for SE colon and rectal cancer showed downward reflections and hence declining annual changes; however, the annual change for 5/1- and 5-year survival for colon cancer showed a small temporary increase after the year 2000 (Figure 4a,d). Survival in SIC remained below colon and rectal cancers.

In Appendix A for 1-year survival, the DK male and female survival for colon cancer in the last period (2016–2020) was better than that for the other countries; FI survival was the worst (for women it was shared with SE). For rectal cancer, DK men and NO women survived the best, and FI men and women survived the worst. For SIC, NO men and FI women survived the best and DK men and women survived the worst.

For 5-year survival in the last period, the ranking was quite similar to 1-year survival (Appendix A). DK men and women survived the best with regard to colon (72.9 and 73.0%) and rectal cancers (72.0 and 74.9%, many of the comparisons to other countries were significant with non-overlapping 95% CIs), while FI survival was the worst, except that SE women survived the worst for rectal cancer. For SIC, NO was the leading country and DK was the last country. In the last period, no significant sex differences were observed, except that FI men survived worse than FI women for colon and rectal cancers. Improvement in 5-year survival of colon cancer over the 50-year period was highest for DK men (37.3% units) and women (34.7% units). For male rectal cancer, NO improved most (37.7% units) while FI showed the highest female improvement (38.1% units). For SIC, SE men (34.1% units) and DK women (37.2% units) showed the best progress.

The 5/1-year survival for colon cancer was also dominated by DK and the result for FI was the weakest (Appendix A). For rectal cancer, survival was the best for DK men and the worst for FI men; for women, FI was the best and SE was the worst. For SIC, FI men and NO women showed the best survival and DK showed the worst survival for men and women.

## 4. Discussion

Improvements in survival in colon and rectal cancers are well-known both internationally and in the Nordic countries [27,28,36,37,38]. Survival data for SIC are less well-known and will be discussed later. The unique feature of the present study is that it covers national survival data of four countries through a half century, which is not possible elsewhere because of a lack of historical cancer registry data. The long observation time could demonstrate that survival improvements have not only been a recent achievement but significant increases had already been gained in the 1970s. Furthermore, the follow-up to the end of 2020 is the most up-to-date according to what national cancer registries can deliver. The present analysis used three metrics of relative survival (1-, 5- and 5/1-year survival) and applied breakpoint analysis and monitoring of annual changes in survival to pinpoint the times when survival changed in the four countries. The main conclusion was that survival patterns were country-specific. For colon cancer, the slopes of survival curves bended upwards for DK, were almost linear for NO and bended downwards for FI and SE. The slope changes were documented as annual changes, which were largest for DK towards the end of the follow-up period; for NO, they were even throughout, and for FI and SE the most positive development was in the early follow-up. For rectal cancer, the patterns were quite similar to those of colon cancer; however, for FI women the slopes were close to linear and for NO women they deviated downwards. DK was the only country for which the 5/1-year survival curve in colon cancer was almost superimposable with the 1-year survival curve, while in NO and SE the 5/1-year curve lagged behind the 1-year curve; thus, DK was effectively improving survival after year 1 of diagnosis, which translates to the steeply increasing 5-year survival. This is the explanation for the DK achievement in 5-year survival over the other countries. A second conclusion was that for rectal cancer, in all countries the 5/1-year curves lagged below the 1-year curves, but DK was able to narrow the gap with time and thus effectively boost 5-year survival. A further conclusion was that while the patterns of the survival curves were country-specific, they were identical for men and women in each country. Female survival for colon and rectal cancer was slightly better than male survival, but the only significant sex difference in 5-year survival in the last period was for FI.

To put the DK survival data into a wider international perspective, a comparison of 5-year survival data with the US Surveillance, Epidemiology and End Results (SEER) program (https://seer.cancer.gov/statfacts/) shows an advantage of the DK data. For 2011–2015, the DK survival for male and female colon cancer was 68.3 and 69.3% compared to the US white population for the year 2014 of 63.8 and 64.7%; for rectal cancer, the rates were 70.3 and 71.9% for DK compared to the US rates of 67.2 and 69.2%. In a study including several countries, 5-year survival for colon cancer in 2010-14 for men and women combined was 70.9% in Australia, 66.8% in Canada, 65.7% in DK and 58.9% in the UK [37]. Survival in rectal cancer was 70.8%, 67.0%, 69.1% and 62.1%, respectively. These cited DK data were 2–3% units below our above data for the one-year-later period (2011–2015). One can calculate from Appendix A that in that period, a one-year increase in DK colon and rectal cancer survival was a remarkable 2 % units.

The reason for the DK top ranking in survival in colon and rectal cancers was not that DK would have started at the high level; DK 5-year survival in these cancers in 1971–1975 was below NO and SE, except that NO male rectal cancer was below that of DK. Thus, DK was able to increase absolute 5-year survival for colon cancer more than the other countries, and for rectal cancer it was second after NO in men and after FI in women. How was DK able to achieve the boost in colon and rectal cancer survival? DK established a national cancer policy in the year 2000, which ensured funding for cancer care and instigated administrative changes for accelerated cancer care pathways [39]. The present data on breakpoints in survival improvement show that the shift to positive development started somewhat before the national cancer policy, for rectal cancer in the early 1990s and for colon cancer in the late 1990s, suggesting that some elements of the positive development were already in place before the inauguration of the national policy. The improvements in health care organization in DK and the other Nordic countries included multidisciplinary treatment planning, centralization of cancer surgery, improved access to endoscopic investigations, increasing use of laparoscopic surgery and improvements in postoperative care [38,40,41]. Even in developed countries, important determinants of survival in cancer are early detection and active treatment of even older patients and of those with comorbidities [42,43]. An earlier study on colorectal cancer survival found that one explanation for poor survival in DK compared to the other Nordic countries was high early mortality (before 6 months) [44]. This was also evident in poor DK performance in 1-year survival between 1980 and 2000 (Appendix A).

Data on survival in SIC are limited and complicated by the almost equal proportions of adenocarcinoma and the indolent NET. Mortality has been mainly due to adenocarcinoma, but survival in this has increased [5,6]. Our results showed that survival in SIC was initially below that of the other cancers but the improvement was rapid and 5-year survival caught up with the other cancers for FI and NO. The success of DK for colon and rectal cancers did not extend to SIC, and the DK results were worse among the Nordic countries. This is curious because the treatment and the treating clinics are most likely shared by SIC and colon and rectal cancers. The DK survival curves for SIC were, however, increasing steepest among the Nordic countries after the year 2000, and DK may be catching up.

As for limitations, NORDCAN is lacking pathological and clinical data which excludes the possibility of considering the stage in survival analysis. However, the comparability of stage data over long time series or between countries is not simple and was not applied in a recent study between the Nordic countries, even though data were available but were collected in different ways [38]. Lacking stage data do not allow for the assessment of the contribution of early detection in increasing survival. However, comparison of 1- and 5/1-year survival allows for the assessment of the death rates between periods 0–1 and 1–5 years after diagnosis. The fact that these curves were close to each other for colon cancer and SIC is an achievement as no more patients were lost in four subsequent years than were lost in the first year. For rectal cancer, more patients were lost in the four subsequent years. To compensate for the limitations, the NORDCAN database can excel with features that enable reliable survival analyses: high coverage of cancers which are up to 95% morphologically verified and minimal loss-to-follow-up [18].

## 5. Conclusions

Although survival in intestinal cancers developed well in all Nordic countries, we observed signs of slowing tempo, particularly in FI and SE, which may signal the limits which can be reached with current forms of awareness, screening, diagnostics and treatment. Earlier studies have reported relative weaknesses in survival outcomes in DK, but the present data show a very strong improvement since the 1990s which has paralleled organizational changes in medical care and financial allocations to care infrastructure. Although DK medicine has always been at a high international level, the present data, documenting excellent achievements in colon and rectal cancer survival in 20 years, should be an encouragement to other countries struggling with survival outcomes.

## Figures and Tables

**Figure 1 cancers-15-00991-f001:**
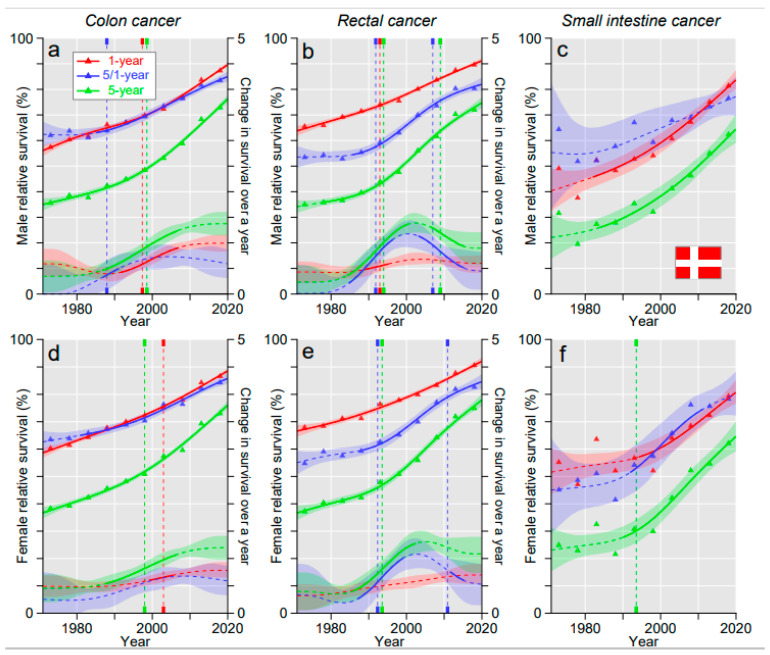
Relative 1-, 5/1- and 5-year survival in Danish men (**a**–**c**) and women (**d**–**f**) in colon (**a**,**d**), rectal (**b**,**e**) and small intestinal cancers (**c**,**f**). The vertical lines show significant breakpoints in survival trends and bottom curves show estimated annual changes in survival. The solid lines in survival curves and annual change curve indicate a plausible trend (see Methods). All curves are color coded (see the insert).

**Figure 2 cancers-15-00991-f002:**
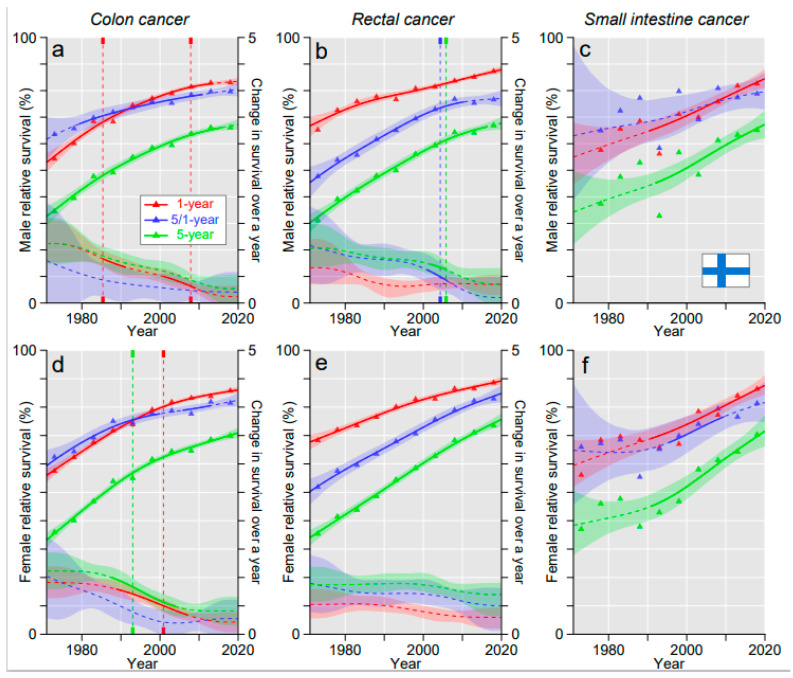
Relative 1-, 5/1- and 5-year survival in Finnish men (**a**–**c**) and women (**d**–**f**) in colon (**a**,**d**), rectal (**b**,**e**) and small intestinal cancers (**c**,**f**). The vertical lines show significant breakpoints in survival trends and bottom curves show estimated annual changes in survival. The solid lines in survival curves and annual change curve indicate a plausible trend (see Methods). All curves are color coded (see the insert).

**Figure 3 cancers-15-00991-f003:**
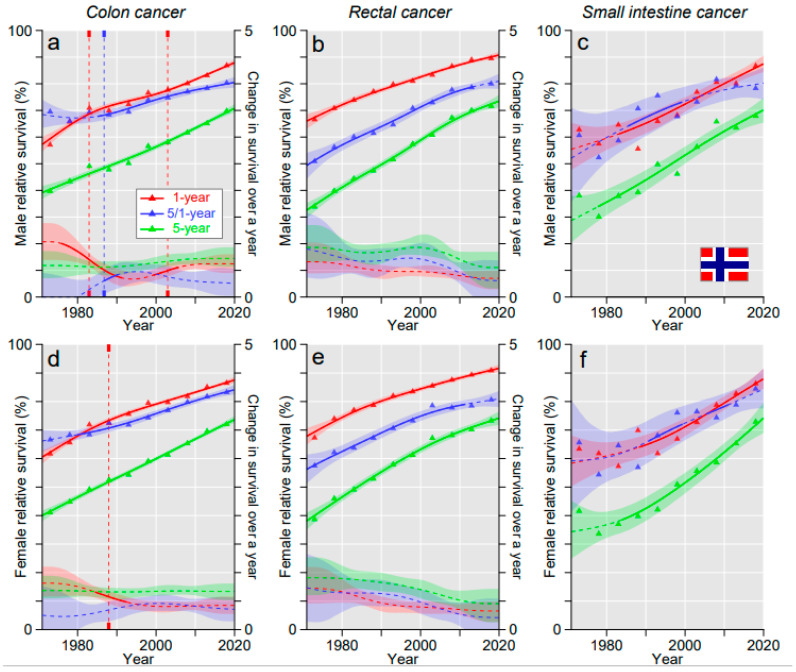
Relative 1-, 5/1- and 5-year survival in Norwegian men (**a**–**c**) and women (**d**–**f**) in colon (**a**,**d**), rectal (**b**,**e**) and small intestinal cancers (**c**,**f**). The vertical lines show significant breakpoints in survival trends and the bottom curves show estimated annual changes in survival. The solid lines in survival curves and annual change curve indicate a plausible trend (see Methods). All curves are color coded (see the insert).

**Figure 4 cancers-15-00991-f004:**
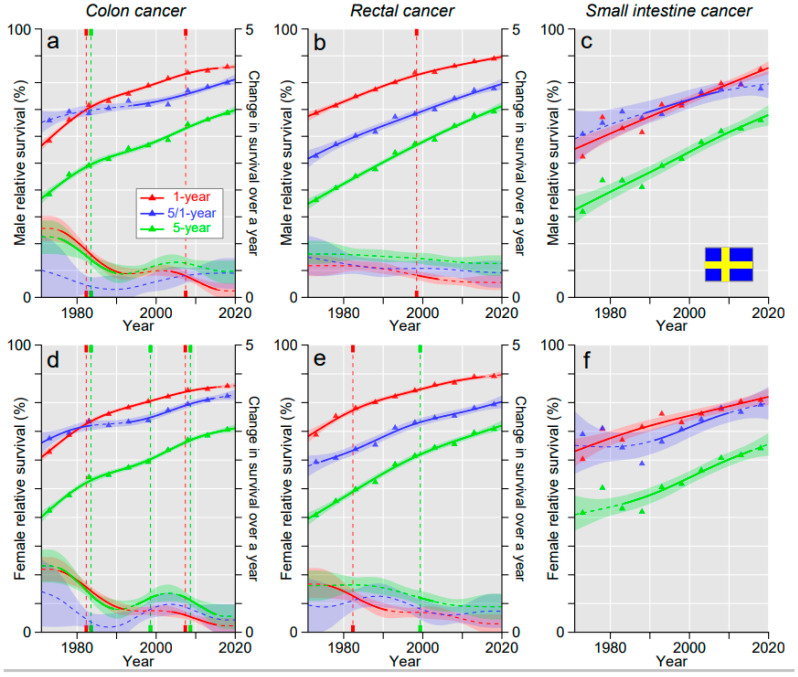
Relative 1-, 5/1- and 5-year survival in Swedish men (**a**–**c**) and women (**d**–**f**) in colon (**a**,**d**), rectal (**b**,**e**) and small intestinal cancers (**c**,**f**). The vertical lines show significant breakpoints in survival trends and bottom curves show estimated annual changes in survival. The solid lines in survival curves and annual change curve indicate a plausible trend (see Methods). All curves are color coded (see the insert).

**Table 1 cancers-15-00991-t001:** Incidence (A) and mortality (B) in colon, rectal and small intestine cancers from 2011 to 2020, separately for males (left part) and females (right part).

(A) Case Numbers, Incidence (ASR) and Cumulative Incidence					
Males	ASR (World)	Cum. Risk % (0–74)	Females	ASR (World)	Cum. Risk % (0–74)
**Colon**					
Denmark, 16,752	27.2	3.2	Denmark, 16,388	23.3	2.7
Finland, 10,311	17.2	1.9	Finland, 10,565	14.3	1.6
Norway, 14041	27.2	3	Norway, 15,330	25.5	2.8
Sweden, 21,576	19.3	2.2	Sweden, 22,122	17.4	1.9
**Rectum**					
Denmark, 9585	16.4	2	Denmark, 6047	9.5	1.1
Finland, 7078	12.2	1.5	Finland, 4851	7.2	0.87
Norway, 7932	16.4	2	Norway, 5440	10.4	1.2
Sweden, 12,548	12.1	1.5	Sweden, 8221	7.3	0.87
**Small intestine**					
Denmark, 1070	1.9	0.24	Denmark, 852	1.4	0.18
Finland, 911	1.7	0.2	Finland, 756	1.2	0.15
Norway, 1106	2.4	0.28	Norway, 780	1.5	0.18
Sweden, 1620	1.6	0.2	Sweden, 1305	1.2	0.15
**(B) Death numbers, mortality (ASR) and cumulative mortality**					
**Colon**					
Denmark, 6639	9.9	0.99	Denmark, 6685	7.8	0.77
Finland, 3936	6.0	0.61	Finland, 4145	4.5	0.47
Norway, 5578	9.9	0.97	Norway, 6099	8.3	0.82
Sweden, 9007	7.2	0.71	Sweden, 9562	6.1	0.61
**Rectum**					
Denmark, 2783	4.3	0.46	Denmark, 1839	2.3	0.24
Finland, 2675	4.2	0.46	Finland, 1821	2.1	0.23
Norway, 2294	4.3	0.47	Norway, 1588	2.4	0.25
Sweden, 4926	4.1	0.45	Sweden, 3457	2.4	0.26
**Small intestine**					
Denmark, 371	0.57	0.07	Denmark, 349	0.47	0.06
Finland, 359	0.57	0.07	Finland, 290	0.34	0.04
Norway, 333	0.61	0.07	Norway, 281	0.46	0.05
Sweden, 646	0.55	0.06	Sweden, 579	0.43	0.05

## Data Availability

Aggregated data from a publicly accessible database were used, posing no ethical issues, and can be accessed at https://nordcan.iarc.fr/en. The full statistical R code is available at https://github.com/filip-tichanek/nord_intestine.

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
