# Peer review of "Survival in Colon, Rectal and Small Intestinal Cancers in the Nordic Countries through a Half Century"

_cancers, 2023, doi:10.3390/cancers15030991_

Round 1
Reviewer 1 Report
Interesting paper. Like all epidemiogical studies there is the possibility of biases and the Authors shoul try to minimize them explaining in detail the methods used fro data collection, the methods used to certify the cause of death.
I would like to suggest, if I am allowed to do so, to include at the end of the paper a subchapter "LIMITATIONS" explaining the possible and potential statistical biases.
Author Response
Reviewer 1
We thank the reviewer for his comments. The methods are described in detail in lines 64-99. We have added also text to Discussion about the safeguards of NORDNET data against survival bias (lines 237-8:
We highlighted the introduction to the paragraph of limitations…l. 212, and added to the end (l. 218-9.
To compensate for the limitations, the NORDCAN database can excel with features that enable reliable survival analyses: high coverage of cancers which are to 95% morphologically verified and minimal loss-to-follow-up (18).
We have also edited the text and made small linguistic changes.
Reviewer 2 Report
Dear authors and editor,
The manuscript titled ,, SURVIVAL IN COLON, RECTAL AND SMALL INTESTINAL CANCERS IN THE NORDIC COUNTRIES THROUGH A HALF CENTURY’’ analyses the relative survival rates for colon, rectal and small intestinal cancers from Denmark, Finland, Norway and Sweden.
This is an interesting project based on a big size population (limited to Nordic countries). The authors found that during the 50 years of observation in the NORDICAN system, the 5-year survival rate generally improved. 5-year and 1-year survival rates vary and may result from the organization of health care system in a particular country. We know from medical data as well as statistical data from the European Commission that Denmark has the best healthcare system in the entire EU. The reviewed manuscript confirms only known data and is limited to Nordic countries population only. In my opinion, the project is not innovative. It duplicates many already published information in this field.
From a technical point of view: the paper is well-organised, the language is correct and the content is understandable. The manuscript written in good scientific language. Literature properly selected.
However the manuscript is good, I have some comments that should be clarified.
1. What is innovative in this paper?
2. What new information does this project bring?
3. From the authors, only one (Akseli Hemminki) has affiliations from the Nordic country whose data is being analysed, so with a high degree of probability only he knows the specifics of the health care system in the analysed countries, but unfortunately he is not the main author as well as the head of the project.
In summary, I support the publication of the manuscript.
Thank you for your choice me as a reviewer.
Author Response
Reviewer 2
We thank the reviewer for his comments and find his point of lacking innovation of challenge.
We added text to the beginning of Discussion (l. 169-172) about innovation/novelty.
The unique feature of the present study is to cover national survival data of four countries through a half century which is not possible elsewhere because of lacking historical cancer registry data. The long observation time could demonstrate that survival improvements were not only a recent achievement but significant increases were gained already in the 1970s. Furthermore, the follow-up to the end of 2020 is most up-to-date what national cancer registries can deliver.
This whole paragraph includes other aspects of novelty, which are then detailed later in Discussion.
Round 2
Reviewer 2 Report
Dear authors ,
Undoubtedly, the corrected manuscript is improved.
The authors partially and superficially answered my questions in the presented version of the manuscript.
In conclusion, I support publication of the presented article.
Thank you